# The Effect of Acidity Coefficient on the Crystallization Properties and Viscosity of Modified Blast Furnace Slag for Mineral Wool Production

**DOI:** 10.3390/ma15134606

**Published:** 2022-06-30

**Authors:** Tielei Tian, Xinyu Jin, Yuzhu Zhang, Yue Long, Xinlin Kou, Jiayi Yang

**Affiliations:** College of Metallurgy and Energy, North China University of Science and Technology, Tangshan 063210, China; jinxinyu@ncst.edu.cn (X.J.); zyz@ncst.edu.cn (Y.Z.); longyue@ncst.edu.cn (Y.L.); kouxinyu@ncst.edu.cn (X.K.); yangjiayi@ncst.edu.cn (J.Y.)

**Keywords:** modified blast furnace slag, iron tailings, crystallization, viscosity, DSC

## Abstract

The crystallization and viscosity of modified blast furnace slag are key factors in fiber forming conditions. In this paper, the crystallization behavior of modified blast furnace slag under continuous cooling conditions was studied by differential scanning calorimetry, and its crystallization kinetics with different acidity coefficients were established. On this basis, the evolution law of the crystallization phase and the influence of crystallization on the viscosity of modified blast furnace slag with different acidity coefficients were analyzed. The results indicated that the crystallization phases of slag with acidity coefficients of 1.05 and 1.20 were, respectively, Melilite and Anorthite. During the cooling process at the acidity coefficient of 1.05, the critical rates of precipitation of Melilite and Anorthite were 50 °C/s and 20 °C/s, respectively, while they were 20 °C/s and 15 °C/s, respectively, at the acidity coefficient of 1.20. With the increase of the acidity coefficient, the crystal growth mode of slag changed from two-dimensional and three-dimensional mixed crystallization to surface nucleation and one-dimensional crystallization. The crystallization activation energy of slag with acidity coefficients of 1.05 and 1.20 were 698.14 kJ/mol and 1292.50 kJ/mol, respectively. In addition, the change trend of viscosity was related to crystal size and content.

## 1. Introduction

The Chinese iron and steel industry is dominated by the blast furnace-basic oxygen furnace long process [1]. However, there are many issues associated with the process, like pollution production links, high energy consumption, complex production system and more. In addition, carbon dioxide emissions per ton of steel are about 1.85–1.87 tons [2,3], accounting for about 10% of total domestic CO_2_ emissions [4]. In this regard, the iron and steel industry can be described as a typical high-carbon emission industry. Therefore, it is very important to reduce emissions and reduce energy consumption in the production process. Molten blast furnace slag is the main by-product of blast furnace ironmaking and its annual output is high, containing a large amount of high-quality sensible heat [5]. The heat released from a ton of slag is equivalent to the heat released from the combustion of 58–60 kg of standard coal [6,7]. Therefore, it is necessary to recycle this energy.

Mineral wool is an excellent material with great heat preservation, which has been widely used in construction, industry and other fields [8,9,10,11]. Traditionally, the production process of mineral wool uses basalt as raw material, which is smelted in cupola, and, then, mineral wool is prepared by centrifugal wire throwing using a high-speed, rotating centrifugal roller. This process consumes about 300–400 kg of coke to produce one ton of mineral wool according to [12], and generates large amounts of pollutants, which seriously pollutes the environment. The components of molten blast furnace slag are very similar to the basalt of rock wool material, so the molten blast furnace slag can replace the basalt to produce mineral wool. This could solve the problems that the cupola smelting process has in terms of its high energy consumption and large pollutant emissions from the sources. This is of great significance to the green and low-carbon production of mineral wool industry and the goal of carbon neutralization and carbon peak.

In China, blast furnace slag is mainly basicity slag which has a low viscosity and is easy to crystallize [13,14], while the appropriate viscosity range in the general melt fibrosis process is 1–3 Pa·s [15]. Obviously, it would be difficult for blast furnace slag to directly produce mineral wool. For the modification necessary for this purpose, iron tailings with a high silicon content should be added to the blast furnace slag. Not only would the sensible heat of the blast furnace slag be fully utilized to dissolve the iron tailings, but also modification of the blast furnace slag would improve the acidity coefficient leading to an improved slag viscosity, surface tension and structure. At the same time, the sensible heat recovery of the blast furnace slag would be realized and high-quality fiber materials prepared.

The key parameters in the fibrosis process of modified slag are viscosity and crystallization, because the viscosity directly affects the fiber length, diameter, toughness, thermal conductivity and stability and other parameters. If the viscosity is too low, slag balls rather than fibers would be obtained in the fibrosis process. The melt crystallization would increase the viscosity of the modified slag. There is a significant relationship between crystallization and viscosity and studying the crystallization behavior of modified blast furnace slag and its influence on viscosity is constructive. At present, there are few reports on crystallization behavior and its influence on the viscosity of modified blast furnace slag based on differential scanning calorimetry (DSC). Reports have mainly focused on the crystallization mechanism and crystallization phase evolution mechanism of blast furnace slag, coal ash slag and alloy slag. Gan et al. [16] studied the crystallization characteristics and kinetics of blast furnace slag during continuous cooling. Xuan et al. [17,18] studied the crystallization characteristics of the SiO_2_-Al_2_O_3_-CaO-Fe_2_O_3_-MgO slag system on DSC, indicating that basicity had a greater impact on the crystallization trend of coal ash slag than that of Si/Al, and established the relationship between crystallization temperature and liquidus temperature. Esfahani et al. [19,20] analyzed the effect of the components on the crystallization behavior of synthetic slag in the CaO-SiO_2_-Al_2_O_3_-MgO system by the single hot thermocouple technique (SHTT) and clarified the law of variations of crystallization phase with composition. Sohn et al. [21] studied the law of the effect of Al_2_O_3_/SiO_2_ on the crystallization mechanism of the CaO-SiO_2_-MgO-Al_2_O_3_ slag system by confocal laser scanning microscopy (CLSM). Accordingly, it can be seen that the above papers have only studied the crystallization process of molten slag, but the effect of crystallization on viscosity has not been studied. Zhang et al. [22] studied the effect of Al_2_O_3_ on viscosity and the crystallization properties of Fe-Ni alloy slag by CLSM and characterized the morphology and composition of the crystallization phase. Wang et al. [15] focused on the effect of CaO on crystallization properties and viscosity of silicomanganese slag during fiber formation. Their study indicated that melt crystallization has a certain influence on melt viscosity. In the paper we here present, DSC was used to analyze the crystallization characteristics of modified blast furnace slag at different cooling rates, and the evolution law of the crystallization phase of modified blast furnace slag was analyzed. On this basis, the crystallization kinetics model was established, and the influence of the law of changes to the relative viscosity of crystallization was clarified.

## 2. Methods

### 2.1. Experimental Materials

In the present study, the blast furnace slag and iron tailings were provided by an iron and steel enterprise in Hebei province. Their main chemical compositions are shown in Table 1. The acidity coefficient of blast furnace slag was adjusted to 1.20 by increasing the proportion of iron tailings. Iron tailings and blast furnace slag were mixed evenly. Then, the mixture was heated to 1500 °C for hot status remelting in a tube furnace. Under a constant temperature for 60 min, the tube furnace was closed for cooling to room temperature for reserve. The main chemical components of modified blast furnace slag are shown in Table 2. The acidity coefficient *M_k_* refers to the mass ratio of acid oxide and basic oxide in raw material compositions, namely:(1)$$M_{k}=\frac{W_{SiO_{2}}+W_{Al_{2}O_{3}}}{W_{CaO}+W_{MgO}}$$

### 2.2. DSC

The crystallization performance of modified blast furnace slag was determined by DSC (Setsys Evolution-Hiden Analytical). First, 20 mg of modified furnace slag was put into the corundum crucible using a small spoon. Then Ar protective gas was turned on. The corundum crucible was put on the support table. It was heated at a rate of 15 °C/min and kept at a constant temperature for 30 min after reaching 1500 °C. Then, it was cooled to room temperature at a certain rate.

### 2.3. Microscopic Detection

First, a 150 g sample of modified blast furnace slag was weighed and put into a graphite crucible. Then, the graphite crucible was put into a tube furnace for heating. The furnace was kept at a constant temperature for 120 min after the temperature rose to 1500 °C. The modified blast furnace slag was fully melted and uniform. Then, the temperature was reduced at a rate of 10 °C/min to 1400 °C, 1300 °C, 1250 °C, 1200 °C and 1150 °C, and kept at a constant temperature for 120 min. Finally, the modified blast furnace slag was quickly taken out and placed in the water and cooled to room temperature. The modified blast furnace slag was analyzed by D/MAX2500PC X-ray for X-ray diffraction (XRD) analysis.

### 2.4. Viscosity Methods

A high temperature viscometer was used to measure the viscosity of the blast furnace slag, and its schematic diagram is shown in Figure 1. First, the modified blast furnace slag was ground to less than 100 mesh with a grinding machine. Then, a graphite crucible containing 140 g of modified blast furnace slag was put into the melt physical property comprehensive apparatus for heating. This apparatus was kept at a constant temperature for 30 min after the temperature rose to 1500 °C to ensure the blast furnace slag was fully melted and uniform. Later, a molybdenum rotor connected with a corundum rod was immersed in the modified blast furnace slag for rotation at 50 r/min. Finally, the melt physical property comprehensive apparatus was cooled at a rate of 3 °C/min. The temperature-viscosity curve was measured. The critical viscosity temperature *T_CV_* is the intersection of two tangents on the viscosity curve. The critical temperature at which the melt viscosity was affected by the crystal was characterized by *T_CV_* [23,24].

## 3. Results and Discussion

### 3.1. Cooling Process Characteristics

As shown in Figure 2, when the cooling rate of slag with acidity coefficient of 1.05 was 10–40 °C/s, the DSC cooling curve had an obvious exothermic peak at 1320 °C–1416 °C. With the increase of the cooling rate, the exothermic peak of the slag gradually moved to the low temperature zone and the peak range gradually became wider. When the cooling rate was 10 °C/s, the DSC cooling curve presented two exothermic peaks. This showed that two kinds of mineral phases were precipitated at a lower cooling rate and the low temperature in the precipitation phase meant crystallization was not easy as the cooling rate increased. The exothermic peaks of slag disappeared when the cooling rate was 50 °C/s, which showed that the blast furnace slag was completely solidified into a glass state.

When the acidity coefficient of blast furnace slag was 1.20 and the cooling rate was 2–15 °C/s, the DSC cooling curve had an obvious exothermic peak. When the cooling rate was 2–5 °C/s, two exothermic peaks appeared on the DSC cooling curve. However, when the cooling rate was greater than 20 °C/s, the slag’s exothermic peaks disappeared. This showed that the crystallization ability of the modified blast furnace slag was evidently reduced with the increase of the acidity coefficient. The increase of acidity coefficient when the modified blast furnace slag was completely solidified into a glassy state meant that the required cooling rate decreased gradually from 50 °C/s to 20 °C/s. The main reason was that, with the increase of acidity coefficient, the viscosity of the modified blast furnace slag increased, which led to a decrease in atomic diffusion ability and inhibited the diffusion of atoms from liquid phase to crystal embryo, which was not conducive to nucleation. As a result, it was not easy to crystallize the modified blast furnace slag at a lower cooling rate. Therefore, blast furnace slag should be modified before centrifugal wire throwing, which could prevent blast furnace slag from generating crystal during cooling, improve the quality of mineral wool fiber and increase the production of mineral wool fiber.

### 3.2. XRD

It can be seen from Figure 3 that when the acidity coefficient of melt was 1.05 and the temperature was cooled to 1450 °C, the slag was completely transformed into a vitreous structure. The melt began to precipitate crystals when the temperature continued to cool down to 1400 °C. The crystallization phase of blast furnace slag was mainly Melilite (Gehlenite and Akermanite) in the temperature range of 1250–1400 °C. However, with further decrease in temperature, the blast furnace slag precipitated a large amount of Melilite, and a small amount of Anorthite began to precipitate at 1150 °C. This showed that Melilite was easier to precipitate than Anorthite.

When the acidity coefficient of melt was 1.20 and the temperature dropped to 1300 °C, the slag was transformed into vitreous structure. The melt began to precipitate crystals when the temperature dropped to 1250 °C. The mineral phase composition was consistent with that of the slag with the acidity coefficient of 1.05. At the same time, a small amount of Anorthite began to precipitate.

In addition, combined with DSC, it could be seen that the precipitated phase in the high temperature zone was Melilite, while the precipitated mineral phase in the low temperature zone was Anorthite. During the cooling, when the acidity coefficient was 1.05, the critical rates of the precipitation of Melilite and Anorthite phases were 50 °C/s and 20 °C/s, respectively, while the critical rates of the precipitation of Melilite and Anorthite phases were 20 °C/s and 15 °C/s, respectively. Therefore, after increasing the acidity coefficient, it was not easy to crystallize the modified slag, which improved fiber fracture performance in the fibrosis process.

### 3.3. Crystallization Kinetics

The Ozawa method [25] and the Kissinger method [26] were used to analyze the crystallization kinetics of the modified blast furnace slag during the cooling crystallization.

According to the classical Kissinger Equation (2), crystal growth activation energy *E*_g_ of blast furnace slag could be obtained.
(2)$$\ln\left(\frac{\beta }{T_{p}{}^{2}}\right)=\ln\left(\frac{AK_{0}R}{E}\right)-\frac{E_{g}}{RT_{p}}$$
wherein *β* means the cooling rate, *T*_p_ means the peak temperature of DSC curve, and *E*_g_ means the apparent activation energy.

A plot of ln(*β*/*T*_p_^2^) versus 1/*T*_p_ according to Equation (2) should yield a straight line; *E*_g_ is obtained from the slope of the line. The crystallization reaction of blast furnace slag was described by *E*_g_.

The crystallization mechanism of blast furnace slag in the cooling process could be judged by the Avrami index *n*. The value of *n* could be obtained by the Qzawa equation
(3)$$\log\left[-\ln\left(1-X\right)\right]=-n\log\beta +n\log\left(T_{0}-T\right)+\log K$$
wherein *X* means the volume fraction of crystallization.

A plot of log[−ln(1 − *X*)] versus log*β* according to Equation (3) should yield a straight line; n is obtained from the slope of the line.

As can be seen from Figure 4, the values of *n* were different but not significant at different temperatures. The average value of *n* was calculated. The value *n* of the blast furnace slag with acidity coefficient of 1.05 was 3.796 and at the acidity coefficient of 1.20 was 1.573. *n* = *m* + *β*, wherein *m* is a factor related to crystal growth and *β* is a factor related to nucleation. Since the crystal nuclei in fibers were mainly formed during the DSC cooling, *β* = 1. Given *n* = *m* + 1, the value *m* at the acidity coefficient of 1.05 was 2.796 and at the acidity coefficient of 1.20 was 0.573, and *m* represented the growth dimension. Therefore, the crystallization mechanism of the blast furnace slag with acidity coefficient of 1.05 was mainly two-dimensional and three-dimensional mixed crystallization. Regarding blast furnace slag having an acidity coefficient of 1.20, its mode of crystallization existed between surface nucleation and one-dimensional crystallization. In addition, with increase of the acidity coefficient, the value of the Avrami index *n* gradually decreased. This showed that the higher the acidity coefficient, the more difficult the crystallization of blast furnace slag was during the cooling.

The apparent activation energy *E*_g_ was calculated by fitting the experimental data, shown in Figure 5. The crystallization activation energy *E* of blast furnace slag with different acidity coefficients was 698.14 kJ/mol and 1292.50 kJ/mol, respectively.

### 3.4. Viscosity

It can be seen from Figure 6 that when the modified blast furnace slag, with acidity coefficient of 1.05, was at 1400–1500 °C, the fluidity of the slag was higher. In this temperature range, with decrease of temperature, the viscosity of the slag changed gently and the stability of the slag was great. However, the change trend of slag viscosity became steeper when the temperature decreased further. Especially when the temperature was lower than 1353 °C, slag viscosity presented abrupt changes and increased rapidly. This was because the slag started to crystallize at 1353 °C, and with the increase and growth of crystals, slag viscosity had an obvious change and the melt eventually lost fluidity.

With the decrease of temperature, the viscosity of the modified blast furnace slag with acidity coefficient of 1.20 gradually went up. However, slag viscosity had no abrupt changes while there were few changes at the beginning of crystal formation. When the melt temperature was lower than the crystallization temperature 1325 °C, the melt was a heterogeneous system composed of liquid and solid mixture. However, slight changes to the liquid phase composition had insignificant effects on the viscosity. The viscosity changed due to crystal content and size, and crystal precipitation was inhibited after the increase of the acidity coefficient. Therefore, the viscosity of the modified blast furnace slag with acidity coefficient of 1.20 changed little with temperature.

In addition, when mineral wool was formed, the viscosity was required to be 1–3 Pa·s. This meant that the appropriate temperature of blast furnace slag fiber formation was 1351–1359 °C and the narrow temperature was 8 °C. The appropriate temperature of the modified blast furnace slag fiber formation was 1288–1397 °C and the wide temperature was 109 °C. The temperature range of direct fiber formation of blast furnace slag was narrow, which was not conductive to the fiber formation rate and stability of the mineral wool fiber. Therefore, an increase of acidity coefficient led to reduced difficulties in mineral wool fiber production process. The quality and yield of mineral wool fiber could also be guaranteed.

## 4. Conclusions

The slag was completely solidified into a glass state when the cooling rate of slag with acidity coefficients of 1.05 and 1.20 were 50 °C/s and 20 °C/s respectively.The crystallization phases of the slag with acidity coefficients of 1.05 and 1.20 were Melilite and Anorthite, respectively. In addition, the critical precipitation rates of Melilite and Anorthite gradually decreased with increase of the acidity coefficient.With the increase of acidity coefficient, the crystal growth mode of slag changed from two-dimensional and three-dimensional mixed crystallization to surface nucleation and one-dimensional crystallization. The activation energies of crystallization of slag with acidity coefficients of 1.05 and 1.20 were 698.14 kJ/mol and 1292.50 kJ/mol, respectively. The strength of crystallization ability could be reflected by the crystallization activation energy. The higher the activation energy, the greater the force required to overcome molecules during crystallization. Therefore, it was more difficult for the slag with acidity coefficient of 1.20 to crystallize.The reasons for the differences in the viscosity change trends of slag with acidity coefficients of 1.05 and 1.20 lay in the crystal size and crystal content. In addition, the modified blast furnace slag with acidity coefficient of 1.20 was more difficult to crystallize and more suitable for fiber forming, than that with acidity coefficient of 1.05, and its fiber formation temperature range was wider at 1288–1397 °C. The wider temperature range could reduce the difficulties in technological operation in the production process of slag cotton and improve the output and stability of the fiber.

## Figures and Tables

**Figure 1 materials-15-04606-f001:**
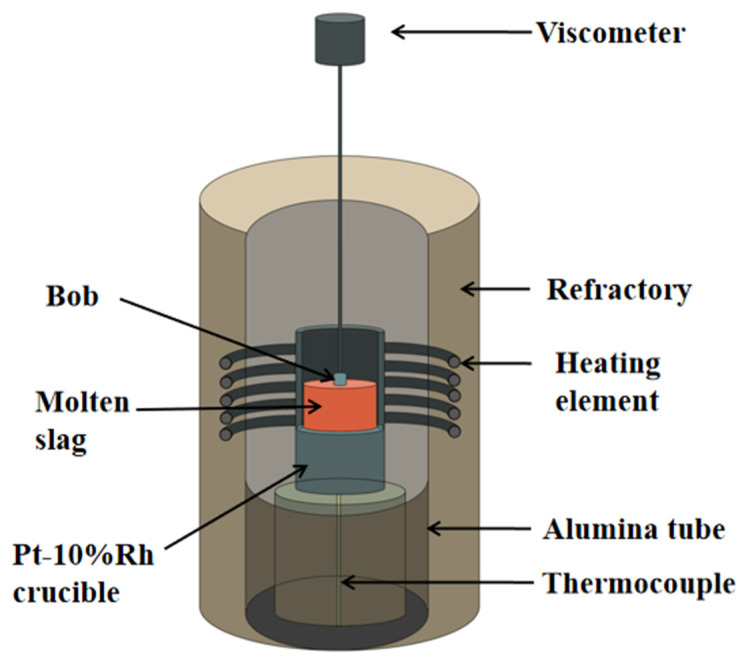
The schematic diagram of the high temperature viscometer.

**Figure 2 materials-15-04606-f002:**
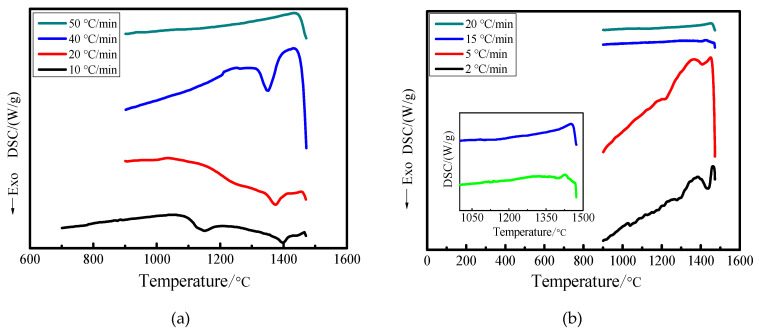
DSC cooling curves with different acidity coefficients. (**a**) Acidity coefficient 1.05; (**b**) Acidity coefficient 1.20.

**Figure 3 materials-15-04606-f003:**
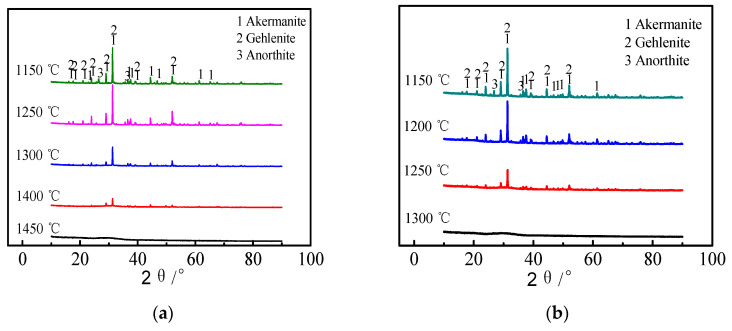
X-ray diffraction of the modified blast furnace slag with different acidity coefficients. (**a**) Acidity coefficient 1.05; (**b**) Acidity coefficient 1.20.

**Figure 4 materials-15-04606-f004:**
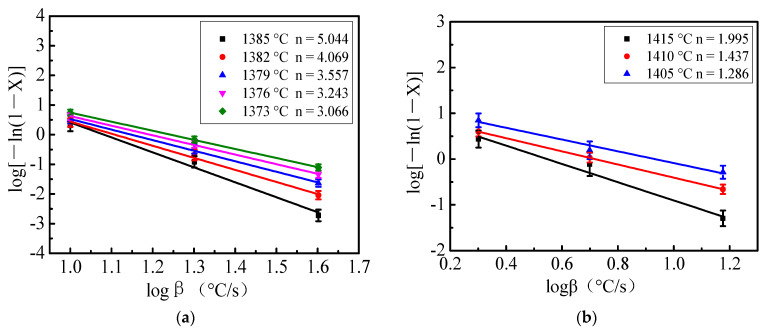
Relationship between logβ and log[−ln(1 − *X*)] of blast furnace slag with different acidity coefficients. (**a**) Acidity coefficient 1.05; (**b**) Acidity coefficient 1.20.

**Figure 5 materials-15-04606-f005:**
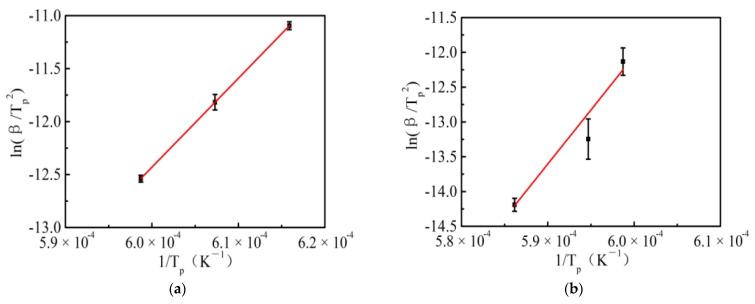
Kissinger’s regression of cooling crystallization kinetics of blast furnace slag with different acidity coefficients. (**a**) Acidity coefficient 1.05; (**b**) Acidity coefficient 1.20.

**Figure 6 materials-15-04606-f006:**
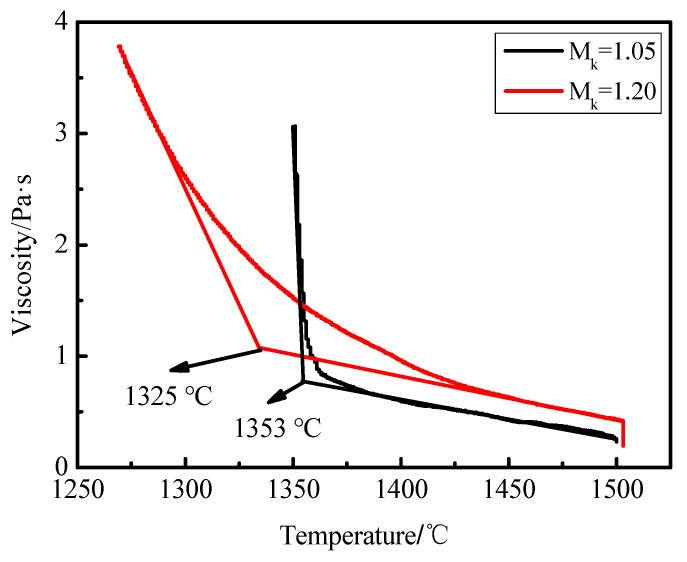
Temperature-viscosity curve of blast furnace slag and modified slag.

**Table 1 materials-15-04606-t001:** Main chemical components of experimental raw materials/%.

Sample	SiO_2_	CaO	MgO	Al_2_O_3_	K_2_O	Na_2_O	Fe_2_O_3_	FeO
BF slag	33.16	37.12	7.96	14.18	0.72	0.45	1.21	0.59
Iron tailings	68.22	8.55	2.20	2.70	3.86	2.28	3.49	4.57

**Table 2 materials-15-04606-t002:** Main chemical components of modified blast furnace slag/%.

Sample	SiO_2_	CaO	MgO	Al_2_O_3_	K_2_O	Na_2_O	Fe_2_O_3_	FeO
*M_k_* = 1.05	33.16	37.12	7.96	14.18	0.72	0.45	1.21	0.59
*M_k_* = 1.20	36.82	34.14	7.36	12.98	1.05	0.64	1.45	1.01

## Data Availability

Data sharing is not applicable.

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
