# Peer review of "The Effect of Acidity Coefficient on the Crystallization Properties and Viscosity of Modified Blast Furnace Slag for Mineral Wool Production"

_materials, 2022, doi:10.3390/ma15134606_

Round 1
Reviewer 1 Report
The manuscript by Tian et al. describes the effect of acidity coefficient on the crystallization properties and viscosity of modified blast furnace slag for mineral wool production.
Considering its application to industry and potential economic, environmental and energy impact, this is important work by Tian et al. However, the manuscript needs complete revision as it cannot be published in the current form. That being said, I support the publication of this manuscript after the authors resolve all outstanding issues.
The manuscript is poorly written (and I do not mean it in terms of English). It has poor organization, it is illogical (and disconnected) at places, and lacks presentation clarity and explanation of what authors aimed to do (and did).
Below I will give major and minor comments.
Major comments:
1. The introduction needs to be revised to achieve better focus and motivation of what the actual aim of the manuscript is.
2. Please use “methods” and then follow up with 2.1; 2.2 etc…
3. The presentation of “Results and discussion” section needs substantial improvement – it is simply not clear. Basically, the reader is left wondering what is the actual aim of the paper – what did the authors do here – that is not clear from how this section is written.
4. Figures are very poor quality and their presentation, legend and caption need to be improved. For example, in the text below Fig.3 , authors state …” As could be seen from the Fig. 3, the values of n were different but not significant at different temperatures.” How is this possible when in Fig.3, authors use completely different parameters for plotting. And this continues throughout the manuscript. Moreover, Fig. 4 (which I will use as an example) does not have error bars. In scientific writing, error bars must be present in figures otherwise figures are useless. So, please fix that issue throughout the text.
5. Define all acronyms (e.g.What is DSC?). Please expand on references.
6. Table 1&2 need better clarification in the caption.
7. In methods, please provide the schematics, picture or diagram of viscosity experiment machine and setup.
…Basically, looking at this manuscript I would be hard pressed to distinguish it from an AI generated word salad that some use to scam journals… Considering all what I said, my review is extremely generous.
Minor comments:
Please have someone do proper English editing.
Reviewer 2 Report
Manuscript ID: materials-1750279
Type of manuscript: Article
Title: The effect of acidity coefficient on the crystallization properties and viscosity of modified blast furnace slag for mineral wool production
Journal: Materials
The Authors present an study about the the crystallization and viscosity of modified blast furnace slag for the “greener” mineral wool production. The crystallization and viscosity of modified blast furnace slag directly affect affects fiber properties. Acidity coefficient as the main parameter is presented as the mass ratio of acid oxide and basic oxide in raw material compositions. Acidity coefficient of 1.20 was more suitable for fiber forming compering to factor 1.05. The work is suitable for publication in the Materials after major revision.
Several comments and suggestions:
-The abbreviation BF-BOF needs to be clarified in the introduction. Similarly, SHTT and CLSM?
-In 3.3 crystallization kinetics please citate Ozawa method and Kissinger method
-to improve the article, I suggest improving the graphical part (if possible, include some SEM photo of crystallization, scheme of viscosity set up, or material production scheme) ,
- Why you only measured at two acidity coefficient values? It may be necessary to expand the range of measurements to find optimal condition.
- my personal opinion is that the article is somewhat flawed. There is a lack of originality or. Clear clarification of originality. Please justify originality.
- The conclusion is flawed. There is a lack of personal thought about research and future prospective in a related field.
Best regards
Reviewer 3 Report
Reviewer’s comments on the manuscript: “The effect of acidity coefficient on the crystallization properties and viscosity of modified blast furnace slag for mineral wool production” written by Tielei Tian, Xinyu Jin, Yuzhu Zhang, Yue Long, Xinlin Kou and Jiayi Yang
The reviewed manuscript presents the crystallization behavior of modified blast furnace slag under continuous cooling conditions studied by DSC. The crystallization kinetics of modified blast furnace slag with different acidity coefficients were also established. In my opinion the manuscript is in the journal’s fields of interests. Moreover, it is interesting and well-written. Experiments are properly planned and the obtained data are clear. Presented discussion is also convincing. It is very good and interesting manuscript. Generally speaking – well done! I recommend to accept it after minor revisions.
Things that should be improved/added before the publication:
- All manuscript: It's probably the template's fault, but often words are not broken down correctly into syllables. Maybe it will be possible to correct it.
- Abstract: abstract should be improved in order to present the relevance of the studies.
- Abstract: please present all numbers with the same number of significant digits.
- Table 1: please present all numbers with the same number of significant digits.
- Fig. 3, Fig. 4: Please attach error bars to the graphs.
- Please add some references after the sentence: “ This process produces per 1 ton of mineral wool by …”
- Authors should decide if they want to make space between number and its unit or not, and then do it consequently.
- References should be carefully checked and corrected.
Round 2
Reviewer 1 Report
I appreciate the improvements to the manuscript. While I was hoping for more extensive improvements and few things remain unresolved, I recommend this manuscript for publication. However, before publication, someone must re-read and edit the manuscript for grammatical errors and typos - there are too many of those for me to list.
Sincerely,
Reynold
Reviewer 2 Report
The manuscript was improved. The authors answered all my questions.
